# Design and Development of an eHealth Service for Collaborative Self-Management among Older Adults with Chronic Diseases: A Theory-Driven User-Centered Approach

**DOI:** 10.3390/ijerph19010391

**Published:** 2021-12-30

**Authors:** Mirjam Ekstedt, Marie Kirsebom, Gunilla Lindqvist, Åsa Kneck, Oscar Frykholm, Maria Flink, Carolina Wannheden

**Affiliations:** 1Department Learning, Informatics, Management and Ethics, Medical Management Centre, Karolinska Institutet, 171 77 Stockholm, Sweden; frykholm@gmail.com; 2Department of Health and Caring Sciences, Faculty of Health and Life Sciences, Linnaeus University, 391 82 Kalmar, Sweden; 3Department of Health and Caring Sciences, Faculty of Health and Life Sciences, Linnaeus University, 351 95 Växjö, Sweden; marie.kirsebom@lnu.se (M.K.); gunilla.lindqvist@lnu.se (G.L.); 4Department of Health Care Sciences, Ersta Sköndal Bräcke University College, Stigbergsgatan 30, Box 111 89, 100 61 Stockholm, Sweden; asa.kneck@esh.se; 5Department of Neurobiology, Care Sciences and Society, Karolinska Institutet, 141 83 Stockholm, Sweden; maria.flink@ki.se; 6Women’s Health and Allied Health Professionals Theme, Karolinska University Hospital, 141 86 Stockholm, Sweden

**Keywords:** eHealth, mHealth, chronic disease, patient activation, person-centered care, self-management

## Abstract

The increasing prevalence of chronic conditions and multimorbidity poses great challenges to healthcare systems. As patients’ engagement in self-managing their chronic conditions becomes increasingly important, eHealth interventions are a promising resource for the provision of adequate and timely support. However, there is inconclusive evidence about how to design eHealth services to meet the complex needs of patients. This study applied an evidence-based and theory-informed user-centered design approach in three phases to identify the needs of older adults and healthcare professionals in the collaborative management of multimorbidity (phase 1), develop an eHealth service to address these needs (phase 2), and test the feasibility and acceptance of the eHealth service in a clinical setting (phase 3). Twenty-two user needs were identified and a web-based application—ePATH (electronic Patient Activation in Treatment at Home)—with separate user interfaces for patients and healthcare professionals was developed. The feasibility study with two nurses and five patients led to a redesign and highlighted the importance of adequately addressing not only varying user needs but also the complex nature of healthcare organizations when implementing new services and processes in chronic care management.

## 1. Introduction

The increasing prevalence of people living with chronic conditions imposes a large burden on healthcare services worldwide [1]. The most common chronic conditions that account for over two thirds of all deaths globally are heart failure, cancer, chronic obstructive pulmonary disease (COPD), and diabetes [2]. While a person’s illness trajectory may start with one chronic condition, multimorbidity increases substantially with age [3]. Healthcare systems that are designed around single diseases pose great challenges for patients with multimorbidity who need to cope with fragmented care services and recommendations from disease-specific guidelines that may be contradictory and cumbersome to comply with [3,4]. Therefore, patients’ engagement in self-managing their condition and maintaining health becomes increasingly important [5].

Successful self-management of chronic conditions requires high poly-literacy (health, medications, and healthcare system), referring to individuals’ ability to take medications as prescribed; manage their symptoms, emotions and lifestyle changes; solve practical problems and cope with the impact of the condition(s) on their daily lives; and know when and how to seek appropriate medical advice when needed [6,7]. Individuals’ ability to reframe their life narrative when living with chronic conditions and their ability to accomplish life and patient work successfully are also central to address the demands that healthcare and life pose [8]. Thus, the planning and implementation of tailored care interventions for people with multimorbidity should consider the differences in patient capacity as well as medical characteristics.

Self-management interventions, across different chronic conditions, can contribute to improved health outcomes [9]. In particular, eHealth interventions, defined as “health services and information delivered or enhanced through the internet and related technologies” [10] (p. 1), have been suggested as a promising resource for the provision of adequate and timely support in the self-management of chronic conditions [11,12] and multimorbidity [13]. A growing body of promising evidence of improved health outcomes and cost-effectiveness from clinical studies supports the use of eHealth services [14,15,16,17]. However, real-world effectiveness depends on patients’ uptake, engagement, and long term adherence to these interventions [18].

Despite the major push to harness eHealth services that support self-management behavior among older adults with chronic conditions in interaction with primary healthcare, precisely how to develop theory- and evidence-informed eHealth interventions that engage users remains a challenge and is rarely well documented [12]. Due to the high prevalence of chronic conditions, older adults represent one of the demographic groups that could benefit most from eHealth services yet they may also be among those who experience most difficulties in using them [19]. The accumulation of treatment and illness burdens associated with aging serve as negative feedback loops that may constrain individuals’ capacities for healthy behavior and sustained self-management [4].

To create self-management support interventions that are meaningful, manageable, and sustainable for the heterogeneous groups of people with multimorbidity, technology needs to be based on knowledge about these groups’ specific needs [7,20,21,22] and constructed in a way that takes the complex nature of managing chronic conditions into account [14]. Thus, the systematic development of complex interventions for specific user groups should be based on the best available evidence and appropriate theory [23,24], following principles of participatory design promoting a common understanding and acknowledging the importance of including all stakeholders’ perspectives during the design process [25]. Therefore, the aim of this study was to apply an evidence-based and theory-informed user-centered design approach for (a) identifying the needs of older adults and healthcare professionals in the collaborative management of multiple chronic conditions, (b) developing an eHealth service to address these needs, and (c) testing the feasibility and acceptance of the eHealth service in a clinical setting.

## 2. Materials and Methods

### 2.1. Design

This study used a theory-driven user-centered design approach, which implies that the design builds on theory and existing knowledge, while also promoting close collaboration between patients with chronic health conditions, healthcare professionals, researchers and service designers to identify needs and develop design solutions [26].

### 2.2. Theoretical Underpinnings

The study design was guided in particular by the Chronic Care Model (CCM) [27,28], the self-determination theory [29], and evidence from a large-scale intervention to improve care transitions [30]. The CCM is a validated multidimensional framework designed to improve care for patients with chronic conditions [27,28]. A critical aspect of the CCM is the close relationship between a prepared and proactive practice team and activated patients and their families as integrated members of the interdisciplinary teams. A further developed framework, the eHealth enhanced eCCM, takes into account the opportunities provided by the use of eHealth [31].

As the adoption of self-management support services is dependent on individuals’ motivation to engage in a change process, we relied on the self-determination theory, which proposes three fundamental and universal psychological needs that are central for behavior change [29]. These needs are autonomy (“feeling of willingness and volition with respect to one’s behaviors”), competence (“feeling effective in one’s interactions with the social environment”) and relatedness (“experiencing others as responsive and sensitive”) [29] (p. 86). Previous research has shown that the satisfaction of these needs can explain health behavior change [32,33] and enhance internalization of health-promoting behavior [34]. Therefore, these needs should be taken into consideration when designing technologies aimed to foster sustained engagement, behavior change, and wellbeing [35].

Given the heterogeneous results from previous eHealth interventions, our study was particularly guided by the findings from a successful large-scale care transition intervention that identified four essential pillars of self-management support that have proven effective in keeping patients out of hospital: (1) being knowledgeable about medications and management of symptoms; (2) understanding and managing a personalized care plan; (3) being knowledgeable about indications that one’s condition is worsening and how to respond; (4) knowing when to seek care and whom to turn to [36,37]. These pillars guided the user-centered design process.

### 2.3. Overview of the User-Centered Design Process

The user-centered design process was carried out in three interconnected phases (Figure 1): (1) exploration of user needs (patients and healthcare professionals); (2) design and development of content and software; (3) testing and redesign. The design process lasted from January 2016 to December 2017 and involved iteration between the three phases until user needs had been adequately explored, described, and addressed. In line with the ISO 9241-210 standard for human-centered design, users were involved throughout the design and development process [38].

### 2.4. Setting and Participants

The study was conducted in a Swedish healthcare setting, covering several levels of care. The settings were chosen based on established collaboration with the researchers and their interest in developmental work. The first step was conducted at a tertiary care heart failure day care clinic. Steps 2–8 were conducted at an integrated care organization, in which hospital care, primary care, and social services are managed in a single organization [39]. At the integrated care organization, a care development leader was designated as a single point of contact for the research group and as a facilitator in planning, recruitment, and data collection. Healthcare professionals from different disciplines and in different positions in the two settings, as well as patients with heart failure, COPD, or diabetes as their main diagnosis, were purposefully recruited to participate (Table 1). All participants received written and verbal information about the study and were informed that participation was voluntary and that they could withdraw at any time.

### 2.5. Phase 1—Exploration of User Needs

The first design phase (steps 1–3) focused on the exploration of user needs and contextual conditions for self-management support.

#### 2.5.1. Step 1—Exploration of Self-Management

One researcher (Å.K.) performed five days of observations at a heart failure day care clinic to gain a broad understanding of how multimorbidity was managed in patient–professional interactions and performed interviews with registered nurses (*n* = 4), patients (*n* = 8), and family caregivers (*n* = 2) who were recruited through the clinic. The interviews and field notes from observations generated rich experience-based data about patients’ capabilities, skills, motivation, and specific support needs with respect to managing symptoms of chronic conditions in everyday life. The interviews were audio-recorded and transcribed verbatim.

#### 2.5.2. Step 2—Exploration of Cross-Organizational Collaboration

A focus group discussion [40] with healthcare managers (*n* = 4), administrators and quality developers (*n* = 3), and frontline staff (*n* = 2) from different units (primary care level, hospital care, and elderly care) of the integrated care organization was performed to explore cross-organizational care coordination between specialist care and primary care for older people with chronic conditions. The focus group discussion was facilitated by O.F., M.F. and Å.K., lasted 120 min, was audio-recorded and transcribed verbatim.

#### 2.5.3. Step 3—Exploration of Work Processes

To further map work processes and contextual conditions for self-management support, two researchers (O.F. and Å.K.) performed individual in-depth interviews with healthcare managers (*n* = 2), administrators and quality developers (*n* = 3), and frontline staff (*n* = 2). The majority had already participated in the focus group discussion in step 2. Thus, these interviews contributed to a more in-depth analysis of the patient journey of chronically ill patients and explored participants’ reflections on organizational conditions and current work processes that could potentially be supported by eHealth. The interviews were audio-recorded, transcribed verbatim, and analyzed thematically [41] together with the data collected in steps 1 and 2. After the inductive analysis, we categorized identified user needs into themes.

### 2.6. Phase 2—Design and Development

The second design phase (steps 4–6) focused on developing an eHealth tool for self-management based on empirical and theoretical assumptions and elicited user needs.

#### 2.6.1. Step 4—Development of Information Content

A design team consisting of researchers (O.F., M.E., Å.K., M.K.), a quality developer (*n* = 1), and district nurses (*n* = 5) from five primary care centers connected to the integrated care organization was established to develop the information and knowledge content of the eHealth service. The design team also had access to an expert panel consisting of a pharmacist, a physician, and a specialist nurse in cardiology. The design team collected relevant patient information pamphlets and brochures available at the primary care centers and met monthly during a 5-month period to adapt self-management information content for use in an eHealth service.

#### 2.6.2. Step 5—Iterative Software Design, Development, and Testing

In parallel with step 4, the researchers collaborated with a systems developer to design the software. Using user-centered design principles [42], design sketches and mockups were developed by an interaction designer in the research group (O.F.) and discussed with the design team in the monthly meetings. Once satisfaction was reached on the low-fi prototypes, a web application was developed. A patient representative with diabetes tested the early prototypes and contributed with advice for refinement and improvements. The district nurses from the design team were educated in using the eHealth service and discussed new work routines.

#### 2.6.3. Step 6—Validation of User Needs and Design Adjustments

A functional version of the prototype was demonstrated by O.F. and discussed in a focus group with healthcare managers (*n* = 5), administrators and quality developers (*n* = 2), and frontline staff (*n* = 2). All but one of the managers (hospital pharmacist) had participated in phase 1 (Table 1).

### 2.7. Phase 3—Testing and Redesign

The third design phase (steps 7–8) focused on testing the feasibility and acceptance of the developed eHealth service in primary care and on redesign.

#### 2.7.1. Step 7—Feasibility Test in Primary Care

All district nurses from the design team were encouraged to use the eHealth service and introduce it to a convenience sample of patients who were willing to participate in a six-week feasibility test. The eHealth service was provided free of charge in the Swedish language and required that users had access to a computer with internet connectivity. A test protocol was developed together with the district nurses detailing how to set up patient accounts with tailored content and how frequently to interact with patients. We performed follow-up interviews with the patients and district nurses who participated.

#### 2.7.2. Step 8—Refinements and Redesign

In this final step, the eHealth service was redesigned to enhance its usability based on experiences from the feasibility study in primary care. The redesign involved improvements in the web application as well as a new design process to develop a complementary mobile application. Thus, an agile development process with a mobile app developer and the web application developer was initiated. One of the authors (C.W.) created design sketches and mockups which were discussed during bi-weekly sprint meetings with the developers as well as a volunteering patient who had not participated in previous design steps.

## 3. Results

### 3.1. Phase 1—Exploration of User Needs

Phase 1 resulted in the specification of twenty-two user needs from the perspectives of patients, family caregivers, and healthcare professionals, which were grouped into five themes (Table 2). Detailed results from the observation studies and interviews [43], as well as the information flow in the chronic care pathway [44], are reported elsewhere.

#### 3.1.1. Theme 1—Diagnosis-Specific Information

Easily accessible information was important to both patients and healthcare professionals. Participants described information needs as being particularly high at the time of diagnosis and emphasized that information needs to be trustworthy, comprehensive, and easily understandable. The information that is provided to patients should be tailored to their specific needs and preferences, taking their psychosocial condition and possible cognitive impairments into account.
*In the best of worlds, there would be systems for carrying information around and not having to have it in your head or on paper slips or something like that … a good anamnesis too, so you’re not, like, starting from scratch.*(Staff)
*But, like, I think this is really exciting, because I … I think that, like, this technology with gathering information, and then it can really be facilitated by so much being gathered there, and I know what it says there, I have access to it.*(Patient)

#### 3.1.2. Theme 2—Medication Management Support

Patients and family caregivers desired clear information and instructions for managing their medications, as well as reminders. Healthcare professionals desired access to updated medication lists across organizational borders. They also wanted to be able to follow patients’ medication adherence and reasons for non-adherence. Furthermore, the ability to monitor intended as well as unintended effects of prescribed medications was highlighted.
*A lot of people have misunderstood the medication list, I think. … or they’ve taken their medications like they should, but haven’t really, like, understood what it’s all about.*(Staff)
*The last time I was admitted to hospital, there was a lady (doctor) there who took care of all that and she had rewritten the medication list in a very clear and simply way, with reasons and causes for the tablets and what they were for. And a list like that, where you get both the regular support for filling up the pill organizers and because you can see that this tablet is for that particular thing.*(Patient)

#### 3.1.3. Theme 3—Self-Management Support

The ability to monitor symptoms over time and get feedback was important to both patients and healthcare professionals. To facilitate their self-management, patients desired self-management guidance through educational material and tailored plans for recommended self-management activities, motivational support, and reminders. Healthcare professionals expressed a need to understand patients’ self-management competence and support needs in order to identify demands that may accumulate over time due to an increasing imbalance between symptom burdens and self-management capacity. They considered older patients with multiple diagnoses and severely affected by functional impairments as being less likely to internalize information about self-management and to actively adopt new behavioural skills.
*So maybe they don’t weigh themselves every day either, so they don’t, like, notice right away when they start to gain weight, it’s just suddenly: “Oh, but now I weigh ten kilos more than I did two weeks ago.” But if they had weighed themselves every day, then maybe they would have noticed that already on day two, maybe. And then they could have gone to the care center and just: “I’ve started to gain weight.”*(Staff)
*So it feels like it kind of depends on when they got the diagnosis. If they got, let’s say heart failure or COPD when they were maybe fifty to sixty, then they know more about it, they’ve had it for a few years and are more familiar with it. But if they get heart failure or COPD when they’re like eighty-two, then it feels like they can’t be bothered to take in that information, it feels like they think like: “But you can solve that.”*(Staff)

Therefore, healthcare professionals desired decision support to be able to recommend proactive symptom-management strategies to sustain wellbeing and independence for patients with multimorbidity. They stressed that simple communication channels between patients and healthcare professionals are needed to be able to assist with customized support and feedback to promote patients’ motivation and skills.
*It really has to be easy to get started, to get into it, of course … I mean, of course, it can’t be anything childish, but I mean, like, something like … “This week you’ve exercised every single day or … seven times, well done!” I think you would see that as something positive, like feedback … I think maybe you could have one of those simple, that you just have like a smiley, instead of having to write.*(Staff)

#### 3.1.4. Theme 4—Care Coordination Support

Patients as well as healthcare professionals emphasized the importance of a mutual understanding of roles and responsibilities. For patients, it was particularly important to have access to their healthcare professionals’ contact details, get support in navigating the healthcare system, and receiving appointment reminders. Similarly, healthcare professionals desired support for coordinating a multi-professional care team, for example, by enabling information exchange between providers and facilitating all members of a multi-professional team to access patients’ health records and get an overview of their care trajectory. They also expressed a need for support in collecting patient preferences.
*That kind of information could be there, that if the home care staff doesn’t show up, you call such and such and …? Yeah, or if you feel uncertain or anything like that, and that there is contact information.*(Staff)

#### 3.1.5. Theme 5—Psychosocial Support

It was important to patients and family caregivers to know that support was available when they needed it. Patients wanted to be able to involve their family caregivers in their interactions with healthcare, for example, by inviting them as users of a prospective eHealth service. They also desired to be able to connect with other patients, both to share experiences and strategies to manage their symptoms and worries.
*The thing you think about most is the enormous difference between living in a home with your wife, active and strong, so strong that she does all the day-to-day stuff. You think about that, that is an enormous difference. All those people who live alone and only have an alarm button. I have a, a safe surrounding, without having to use, use an alarm button. Now that is the big difference.*(Patient)
*If you think like this, that I know … there were next-of-kin, if there had been written information, then maybe that could have been shared with them … by e-mail or like … or through some other system … that they are included in this too and can support her in it.*(Staff)

### 3.2. Phase 2—Design and Development

#### 3.2.1. Information Content

There was limited preexisting patient information available at the integrated care organization to guide patients in their self-management. Therefore, new content was developed based on the scientific literature, clinical practice guidelines, and consultations with the expert panel. The design team developed a knowledge base consisting of information and recommendations to support the self-management of type 2 diabetes, heart failure, and COPD. The knowledge base contained: a collection of short diagnosis-specific informative texts covering topics related to symptoms, treatment, and self-management; common diagnosis-specific medications with easily understandable descriptions of their purpose, mechanisms, common side effects and other details; instructions for self-management activities that could be performed at home (e.g., physical activity or diet plans); instructions for the self-monitoring of health parameters (symptoms and outcomes) that would be useful for follow-up.

#### 3.2.2. eHealth Service

In parallel with the development of information content, an iterative software development process resulted in a web-based application—ePATH (electronic Patient Activation in Treatment at Home)—with separate user interfaces for patients and healthcare professionals. Secure login was implemented using e-identification services approved by the Swedish agency for digital government (BankID for patients; SITHS for nurses). Below, we describe the core modules of ePATH (Figure 2a).

#### 3.2.3. Templates Module

Using the templates module, healthcare professionals initialized the ePATH service for patients by setting up individual patient accounts with tailored diagnosis-specific information and self-management recommendations (user needs themes 1 and 3). This was done at an “onboarding” meeting in collaboration with the patients. The content to share with patients was selected from diagnosis-specific templates that contained information content from the knowledge base (i.e., diagnosis-specific information, self-management and self-monitoring recommendations, and medications). A minimal amount of content that was considered relevant for most patients was preselected in the template. Using a toggle function, healthcare professionals could easily select additional content that would be relevant to some patients but not all (e.g., support for tobacco cessation or diet recommendations). The content could also be edited to further tailor support to patients, for example by editing informational texts or fine-tuning the type, intensity and timing of recommended self-management activities and self-monitoring parameters to individual patients’ needs and preferences. After initializing a patient account, healthcare professionals could use the templates module at any time to complement or refine the self-management support content for individual patients.

#### 3.2.4. Information Module

After initialization of a patient’s ePATH account, tailored diagnosis-specific information was accessible in ePATH’s information module. The information module was intended to function as a knowledge base that patients could turn to for learning about their chronic conditions and improving their self-management competence and skills (user needs theme 1).

#### 3.2.5. Interactive Self-Management Modules

Three interactive self-management modules were developed to support patients in their self-care and assist healthcare professionals in tailoring support to patients’ needs and preferences and surveil patient-reported self-monitoring data. Both user groups had the same rights to add, remove, and edit contents. In the patient interface, a self-monitoring feature enabled patients to track and report their medication adherence, self-management performance and health assessments. All tracked data were instantly accessible to patients and healthcare professionals and could be viewed in both tabular and graphical format.

The medications module enabled patients to add their medication list to their ePATH profile to get support in taking their medications and tracking adherence (user needs theme 2). Patients could select medications from the knowledge base and edit details according to their prescriptions (e.g., dosing, strength, administration timing and frequency). Medications that were not available in the knowledge base could be added manually. To receive reminders and report medication adherence, patients had to add details about the prescribed timing for medication intake. Adherence was reported on a binary scale (medication taken/not taken) for each medication administration (i.e., according to prescribed frequency), with an optional free text comment to report reasons for non-adherence.

The self-management activities module contained instructions for self-management activities initially provided by healthcare professionals (user needs theme 3). The type, intensity, and frequency of activities could be edited to adjust to individual needs and preferences. Furthermore, patients could edit which parameters they wanted to track for monitoring performance.

The health assessments module provided instructions for how to self-assess symptoms and outcomes. Although an initial recommendation was provided by healthcare professionals, patients had the opportunity to tailor the parameters to track as well as monitoring frequency.

#### 3.2.6. Messaging Module

The messaging module was designed to enable patients and their assigned healthcare contacts to exchange free text messages with each other, facilitating the exchange of health-related information (user need theme 3) and the provision of psychosocial support (user need theme 5). The ePATH system imposed no restrictions on the number, length, or content of messages. Users could be notified of new messages by email. We anticipated that the messaging module could contribute to strengthen patient’s psychological needs for competence (through the provision of feedback) and relatedness (through the interaction with healthcare professionals).

#### 3.2.7. Care Planning and Coordination Modules

Three different modules were designed to support user needs related to care planning and coordination (user need theme 4), contributing to informed and activated patients in productive interaction with a prepared and proactive practice team. A calendar module was developed to enable both patients and healthcare professionals to get an overview of the patients’ planned self-management activities. Furthermore, patients had access to a contacts module that provided them with an overview of their healthcare contacts, along with descriptions of their roles, responsibilities, and contact information. To support patients in preparing for healthcare visits, an about me module provided a private space for patients to specify goals, diary notes, and memos for personal use.

### 3.3. Phase 3—Testing and Redesign

During a 6-month period, two of the district nurses from the design team and five patients participated in testing the ePATH service. The participants were positive about the conceptual design of ePATH but raised some usability issues that were addressed through redesign. In particular, they experienced challenges gaining an overview of all self-management tasks, and self-tracking required too much effort as patients had to navigate between different pages in the web application. More detailed results from the qualitative analysis will be published elsewhere.

A redesign was made to refine the presentation of content and simplify the interaction with ePATH, e.g., by reducing the number of clicks necessary. The user interface for patients was complemented with a daily overview module that provided an overview of all self-management tasks for the day and simplified tracking (Figure 3). Furthermore, a mobile application (mPATH) was developed to make it possible for patients to get push notifications (if desired), as well as track their self-management and exchange messages with healthcare professionals from a mobile device without logging in to the web application (Figure 2b and Figure 4a). To further reduce the amount of manual data input and clicks, a speed-tracking functionality was developed which made it possible to set default values for selected self-management tasks. For example, an aspired value could be set for a health assessment (e.g., weight = 75 kg) or a self-management activity (e.g., type of physical activity = cycling, duration = 30 min, and intensity = medium). Whenever the aspired value was met, tracking was performed by ticking a checkbox without additional data input. Only deviations from planned self-management activities and anticipated health parameters required manual editing (Figure 4c).

## 4. Discussion

This study applied a theory- and evidence-driven user-centered design process to develop a web- and mobile-based eHealth service for supporting the self-management of chronic conditions among older adults in an integrated care setting. The user needs that we identified concerned a core component of the CCM, namely, the facilitation of productive interactions between informed, activated patients, and prepared, proactive healthcare professionals [27]. Patients as well as healthcare professionals valued easily accessible information, support for medication management, self-management, care coordination, and psychosocial support. While the designed eHealth service addressed identified user needs and organizational considerations, only a small number of participants could be recruited for testing the service in a clinical setting. This may reveal a mismatch between identified needs and readiness to change. Our findings confirm Wagner’s observation that over two decades after the introduction of the CCM, “helping busy practices to transform into effective care systems still remains a formidable challenge” [45] (p. 663).

### 4.1. User-Centered Design Process

We collaborated with patients and healthcare professionals in the user-centered design and development process. It is widely acknowledged that user involvement in the design of information technology is central to a system’s success [46], in particular in terms of user satisfaction and system use [47]. However, user involvement is a double-edged sword that needs to be managed carefully [48]. Challenges (e.g., time constraints, budget, lack of top management support, lack of motivation) need to be considered and there is no clear evidence regarding the optimal degree of user involvement or the stage in the system development lifecycle in which user involvement is most effective. A review by Fischer et al. [49] found that older adults are commonly involved at a low level as informants, testers, and consultants but less frequently as co-designers in the actual design and development process. Similarly, in our study, patients were mainly involved in the exploration of user needs and usability testing. In the design and development process, we involved healthcare professionals as frontline users, primarily to ensure a fit between the eHealth service and work processes.

In the first phase of the user-centered design process, we identified a comprehensive list of user needs and organizational considerations. Consistent with the CCM and previous evidence from the care transition intervention that informed this study [27,30], participants expressed a need for support services that would enable patients to take a more active role in their self-care and facilitate collaboration with healthcare professionals through information exchange. However, we also found that there was variation in patients’ support needs depending on their health status. Whereas some patients are well-off can and want to be in charge of their own care, others are vulnerable and in greater need of support [50]. The timing for support was also important. While information needs were high at the time of diagnosis, healthcare professionals suggested that patients who had lived with their condition for a long time needed more support as they were less able to acquire new knowledge and skills. The need for high-quality support increases in parallel with the accumulation of various factors that negatively affect an individual’s ability or motivation for self-management, such as lack of health literacy, isolation, poverty, distance from healthcare, and absence of social support [51]. Therefore, previous research has emphasized the need to better understand patients’ capacities to manage the burdens of illness and treatment [52] and called for minimally disruptive interventions oriented explicitly towards care for patients with complex conditions [53,54]. Given the extremely heterogeneous group of older adults and the complexity of the illness burden in people with multiple chronic conditions, the importance of personalized care is at its peak.

In the second phase of the user-centered design process, we co-designed and developed self-management support modules that could be tailored to the needs and preferences of individual patients. The developed modules correspond well with self-management components from the PRISMS taxonomy of self-management support that have been identified as suitable for eHealth, namely: patient education and information provision; remote monitoring with feedback and action plans; eHealth-facilitated clinical review; adherence support; psychological support; and lifestyle interventions [12,55]. It has been suggested that the design of self-management support features needs to be aligned with varying user needs and preferences based on patients’ medical and social complexities, demographic factors, readiness for change, and motivation [56,57]. However, there is no conclusive evidence yet about which eHealth components and design features are most effective to support self-management of long-term conditions [12,58]. Thus, more research is needed to uncover relationships between the context, mechanisms, and self-management outcomes related to eHealth interventions [59]. In particular, the question of how to design eHealth interventions that support personalized care merits further investigation.

We specifically aimed to support the personalization of self-management support features to satisfy patients’ psychological needs of autonomy, competence, and relatedness, based on the self-determination theory [29]: autonomy was supported by enabling the tailoring of content; competence was supported by the provision of tailored educational material and by means of self-tracking functionality; relatedness was supported by enabling patients to contact their nurses and get psychosocial support. Personalization is a design feature that offers personalized content or services, which is commonly used in persuasive systems design, where the aim is to promote the reinforcement, change, or shaping of attitudes and/or behaviors [60]. In combination with self-tracking, individual goal-setting and personalized feedback are commonly used personalization features to promote lifestyle changes, whereas it is less common to provide users with the ability to set technical features, such as layouts or prompts [61]. In the self-management modules developed in this study, personalization was not an automated feature provided by the eHealth service. Rather, it was dependent on users’ own settings of technical features and content adjustments. We acknowledge that this requires both health and eHealth literacy and may have influenced ease of use. Previous research shows that eHealth literacy, although an important predictor of eHealth use, is often overlooked in the design of eHealth services for socially disadvantaged groups, such as older adults [62]. It should be considered that physical frailty status per se is negatively associated with older adults’ ICT use independent of age, education, and opinions about the usefulness and usability of ICT [63]. Furthermore, it should be considered that while an eHealth service may be experienced as satisfying or frustrating at the user interface level, behaviors and life in general may be influenced in other ways [35,64].

In the third phase of the user-centered design process, the feasibility and acceptance of the eHealth service was tested in a natural setting and refined. The feasibility test proved challenging, which is reflected in the low number of participants. Only two of the district nurses from the design team participated. The other three were not available for participation due to changes in staffing. Technically challenged staff, resistance to change, age, and eHealth literacy are among the most common reported barriers to the adoption of telemedicine worldwide [65]. Adoption problems may also reflect a mismatch of eHealth services with peoples’ routines in their daily lives [20]. These may be factors that challenged the recruitment of participants for the feasibility test. The nurses acted as interventionists by being tasked with introducing the eHealth service to patients. This contrasts with findings from a systematic review that studies of eHealth tools to support self-management among vulnerable patients with chronic conditions often engage external intervention providers [59]. We believe that one possible explanation for the poor recruitment of patient participants may be that the nurses did not feel sufficiently confident in introducing the tool to their patients, possibly due to insufficient training. A recent systematic review suggests that successful implementation strategies of eHealth programs for patients with chronic conditions living at home include internal and external facilitation, audit and feedback, management support, and training of clinicians [66]. Although training and support was provided and despite the high engagement in the design process and the expressions of confidence and motivation to use the eHealth service in their daily practice, we may have underestimated the effort required by nurses to change existing routines.

### 4.2. Methodological Considerations

A limitation of our study was that it coincided with other development projects, reorganization, and staff leave at the integrated care organization that was the setting for the study. Due to down-prioritization of our project, the process from initial conceptualization of the project to feasibility testing became too long drawn out for the participants to get a return on their investment of time and effort. Despite the strong commitment among participants, including care developers and managers, to contribute to the improvement of self-management support in chronic care, the organizational changes were out of their control. In their review of technology implementation frameworks in health and social care, Greenhalgh et al. [67] found that surprisingly few frameworks considered the organizational setting or the extent of implementation work required. They present a nonadoption, abandonment, scale-up, spread, and sustainability (NASSS) framework that helps technology and service designers in assessing and addressing complexities in the implementation of technologies in healthcare [67]. When reflecting on our study based on their proposed framework, we acknowledge that apart from the eHealth service that was developed, the clinical conditions addressed, the heterogeneous group of patients, and the clinical setting were characterized by high complexity, which is a substantial risk factor for non-adoption or abandonment [67]. Thus, although a user-centered design process that is based on theory and evidence may contribute to successful implementation, the implementation process needs to account for the multiple complexities in the healthcare system, in which unpredictability is ever present [68]. The challenges we experienced may reflect a lack of sufficient managerial support and resources for implementation, which are common challenges in the implementation of eHealth services for self-management [69]. After a lengthy recruitment process, we nevertheless succeeded in performing a small-scale feasibility test with a few nurses and patients from different healthcare organizations. Although our data limit us in drawing conclusions about the usability or usefulness of the developed eHealth service in a real-world setting, we believe that the description of our design and development process may be of value to researchers who plan similar projects. We believe that the user needs that we identified will be applicable in other chronic care settings. However, we acknowledge that older adults are an extremely heterogeneous group with highly varied characteristics and needs, who use, modify, and interact with technologies in rather diverse ways [70]. The merits of user involvement include learning, adjusted designs, and achieving a sense of participation among older users [49]. How far this translates into increased viability of the designed products in the everyday lives of older adults, or even their acceptability, remains an open question for further empirical inquiry.

## 5. Conclusions

Our study has identified needs related to the self-management of chronic conditions and productive interactions between patients and healthcare professionals that could be supported by means of eHealth services. The user-centered design and development process gave insight into the variation of needs and preferences within and between end users that call for personalized support services that can be adapted over time. The ePATH service that was developed could only be tested in a small-scale feasibility study, which limits our ability to draw conclusions about its design features. However, the challenges we experienced in the lengthy process from design and development to finally testing the service highlighted the importance of adequately addressing not only varying user needs but also the complex nature of healthcare organizations when implementing new services and processes in chronic care management. We conclude that user engagement in design and development should not be limited to the elicitation of user needs and creating new services but should focus more holistically on improving current practices to shape better care, which requires adequate strategies and resources to implement changes in people’s lives and complex organizations. We encourage future studies that will further explore how eHealth services can best be designed and implemented to provide personalized support that will meet the varying needs and preferences of persons suffering from chronic conditions and made to fit into the organizational context of healthcare.

## Figures and Tables

**Figure 1 ijerph-19-00391-f001:**
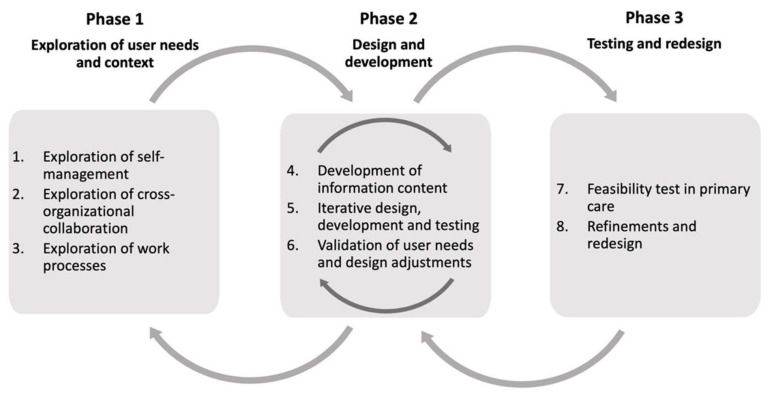
Overview of the three study phases and steps numbered 1–8. The arrows between and within design phases illustrate iteration in the design process. In particular, steps 6 and 8 link the three phases through iteration.

**Figure 2 ijerph-19-00391-f002:**
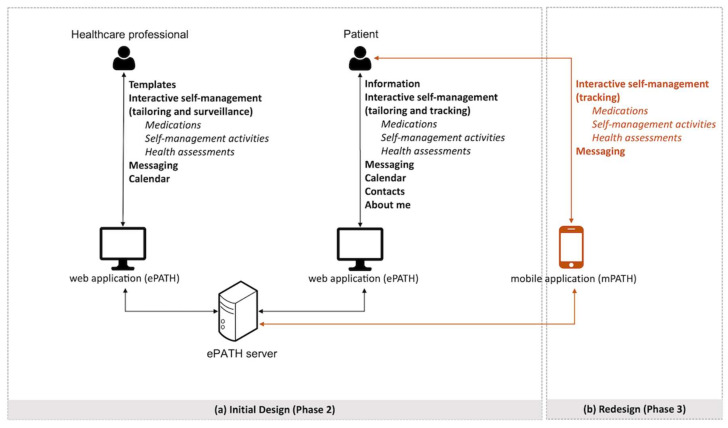
Overview of ePATH architecture and modules for different users: (**a**) illustrates the initial design (phase 2), which enabled all users to access ePATH through a web-based application; (**b**) illustrates how the web-based application was complemented with a patient-facing mobile application and related functionalities for patients after redesign (phase 3).

**Figure 3 ijerph-19-00391-f003:**
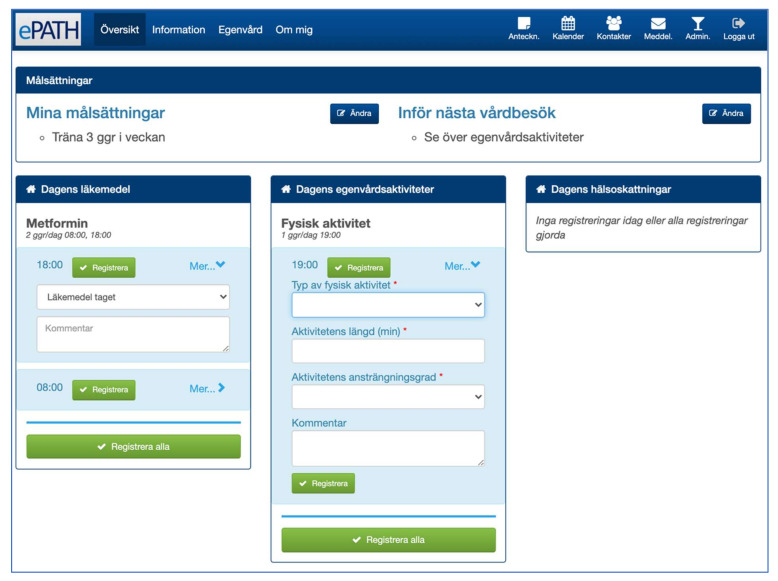
Screen print of the ePATH daily overview module (patient interface): The horizontal navigation panel at the top enables the patient to navigate between different modules (overview, information, self-care, about me, notes, calendar, contacts, messaging). The horizontal panel below (“Målsättningar” = Goals) enables the patient to add general goals and specific goals until the next care visit. The three vertical panels (“Dagens läkemedel” = Medications, “Dagens egenvårdsaktiviteter” = Self-care activities, “Dagens hälsoskattningar” = Health assessments) provide an overview of the planned self-care tasks for the day and enable tracking.

**Figure 4 ijerph-19-00391-f004:**
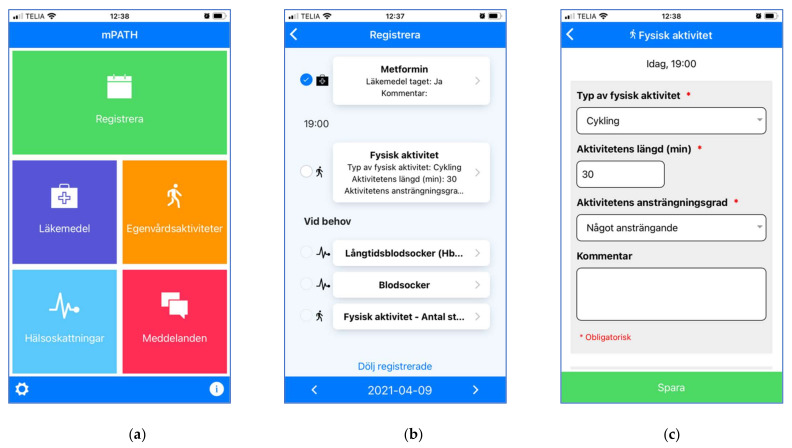
Screen prints of the mPATH app: (**a**) shows the mPATH home page that allows users to navigate to different pages: “Registrera” = Tracking, “Läkemedel” = Medications, “Egenvårdsaktiviteter” = Self-management activities, “Hälsoskattningar“ = Health assessments, and “Meddelanden” = Messaging; (**b**) illustrates an example from the tracking page which lists all daily self-management tasks chronologically. If speed-tracking is possible (first two items in the list), default values are spelled out (“Läkemedel taget: Ja” = Medication taken: Yes) and users can simply click the checkbox to track the task as done; (**c**) illustrates the editing view where the user can edit details for individual self-management tasks.

**Table 1 ijerph-19-00391-t001:** Overview of participant involvement in all design phases, steps 1–8.

			Design Steps							
ID	Participant Role	Organization and Level of Care	1	2	3	4	5	6	7	8
**Healthcare management**										
1	Occupational therapist, Manager	ICO, home care		•	•			•		
2	Registered nurse, Manager	ICO, hospital care		•	•			•		
3	Physician, Manager	ICO, hospital care		•				•		
4	Pharmacist, Manager	ICO, hospital care				•		•		
5	Registered nurse, Manager	ICO, primary care		•				•		
**Healthcare administration & quality development**										
6	Administrator, Social worker	ICC, social service		•	•			•		
7	Quality developer, Registered nurse	ICO, hospital care		•	•	•				
8	Quality developer, Social worker	ICC, social service		•				•		
9	Administrator/Coordinator	ICO, hospital care			•					
**Healthcare staff, frontline**										
10	Specialist nurse, cardiology	ICO, hospital care		•	•	•		•		
11	Physician, internal medicine	ICO, hospital care				•				
12	Specialist nurse, oncology	ICO, hospital care		•				•		
13	Registered nurse	ICO, hospital acute care			•					
14–15	Registered nurses (*n* = 2)	HDC	•							
16–17	Assistant nurses (*n* = 2)	HDC	•							
18–19	District nurses (*n* = 2)	ICO, primary care				•	•		•	
20–22	District nurses (*n* = 3)	ICO, primary care				•	•			
**Patients & family carers**										
23–24	Family carers (*n* = 2)	HDC	•							
25–32	Patients, HF/T2D/COPD (*n* = 8)	HDC	•							
33	Patient, T2D	Personal contact					•			
34–35	Patients, T2D (*n* = 2)	ICO, primary care							•	
36–38	Patients, HF/COPD (*n* = 3)	ICO, primary care							•	
39	Patient, prostate cancer	Personal contact								•

ICO = integrated care organization; HDC = heart failure day care; HF = heart failure; COPD = chronic obstructive pulmonary disease; T2D = type 2 diabetes. •: indicates in which design steps participants were involved.

**Table 2 ijerph-19-00391-t002:** Identified user needs grouped into five themes. Steps 1–3 indicate in which data collection steps the needs were captured.

		Step		
Theme	User Need	1	2	3
1. Diagnosis-specific information	1.1 Easily accessible information	•		•
	1.2 Trustworthy (evidence-based) information	•	•	•
	1.3 Comprehensive information	•	•	•
	1.4 Understandable information	•	•	
	1.5 Information tailored to individual needs	•	•	
2. Medication management support	2.1 Individualized medication management instructions	•	•	
	2.2 Medication reminders	•	•	
	2.3 Access to updated medication lists		•	•
	2.4 Medication adherence and reasons for non-adherence		•	•
	2.5 Monitoring of intended and unintended effects			•
3. Self-management support	3.1 Monitoring of symptoms and wellbeing	•	•	•
	3.2 Support for providing tailored guidance	•	•	•
	3.3 Reminders and motivational support	•		•
	3.4 Information exchange between patients and HCPs		•	•
4. Care coordination support	4.1 Clarification of roles, responsibilities and contact details	•	•	
	4.2 Appointment reminders for patients	•		
	4.3 Overview of patients’ care plan and trajectory	•	•	•
	4.4 Support for collecting patient preferences		•	
	* 4.5 Information exchange between providers			•
5. Psychosocial support	5.1 Assurance of available support	•		
	* 5.2 Support for connecting with other patients	•		
	* 5.3 Support for inviting family caregivers as users		•	•

HCPs: health care professionals; •: indicates in which design steps (1-3) the user needs were identified.; *: user needs that were not addressed in phases 2 and 3 of the design process.

## Data Availability

The data presented in this study are available on request from the corresponding authors. The data are not publicly available to protect participants’ privacy.

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
