# Peer review of "Design and Development of an eHealth Service for Collaborative Self-Management among Older Adults with Chronic Diseases: A Theory-Driven User-Centered Approach"

_ijerph, 2021, doi:10.3390/ijerph19010391_

Round 1

Reviewer 1 Report

The manuscript is focused on the development of an eHealth service, as a request to the increasing multimorbidity and chronic conditions and the capacity of the patient to carry out self-management of their conditions. The also include a pilot study in which they test the developed eHealth service on providers and patients, tailored and personalized to the specific need of the patient.

General comment: The manuscript is well written, easy to follow and understand. However, I suggest trying to shorten the length of the manuscript. Please, try to summarize as appropriate the introduction,

Abstract:

As a personal preference, I would recommend replacing the term “illness” with “pathologies”, which in my opinion is a broad category which involve not only what we define illness (cancers, painful conditions, …) but also chronic systemic conditions, maybe not necessarily defined illness (e.g., psychological conditions)

Introduction:

I would suggest moving the section 1.1 as part of the discussion. Indeed, the introduction is already too long, and by adding also that section, the reader is discouraged from continuing reading.

Materials:

How were the nurses / staff / managers and patients chosen? Voluntary response? Other methodology? Please, specify. Moreover, how were the clinics where the feasibility study was conducted chosen?

It is not clear in Step 7 (line 241) if the eHealth service software was given to the patients or to the nurses. Please, clarify.

Please, provide the age and the gender of patients and providers and staff that performed the pilot study (table 1).

Results:

I would avoid reporting the exact words of patients and healthcare providers within the manuscript, especially for the sake of the length of the manuscript. I suggest attaching as supplemental file and it is up to the reader, if interested, to look for further documentation.

- line 413: I would replace “illness” with “conditions”

- line 418: replace self-management with self-care for the sake of non-repetition of the same word closely in the same sentence.

- line 431: how often was the adherence check? If the patient is supposed to take a medication everyday, as often occurs with chronic conditions, is he/she expected to indicate his/her adherence everyday? Please, clarify the concept of the adherence in terms of frequency, if appropriate.  

- line 465: I believe the first two sentences on the difficulties fit the discussion better. I suggest moving those two initial sentences.

Discussion:

- line 511: why was testing the service harder than expected? If you are referring to the difficulties mentioned in line 465, I suggest moving here those sentences, or to move it to the paragraph from line 590

I would rename the section of Limitations as Strength and Limitations, as in the section you also identified strength of the study.

References:

- please, put in capital letter the journal of BMJ in the references n 26-27. 

I do have some questions that I invite the authors to clarify by adding this information in the discussion.

Is this software going to be free of charge for the patient? Is this expected to be utilized by specific providers or by the entire hospital / healthcare system?

- does the patient need to have a smartphone to allow this software to work? Please, clarify.

- is this going to be offer as standard routine of care or depending on the preference of patient and/or provider?

  • is the software only in Swedish language?

Thank you and looking forward for this corrections. 

Reviewer 2 Report

First of all, I would like to thank the authors for providing the possibility to review their manuscript. Further, I would like to point out, that the manuscript was very well written and clearly structured. For me, as a reader, it was at any time clear to see and follow the structure of the manuscript. In this context, the authors also succeeded in describing WHY they did specific things, or what their thought process was.

However, when reading the manuscript, I stumbled upon some minor issues:

  • Sometimes, there are “ “ (blanks) missing (i.e., line 74, after [11])
  • In Line 152 the authors describe that the design process phase lasted from Januar 2016 to December 2017. Why is the manuscript just now published? Because that is another 4 years. Can the authors provide more information on that? Did the development and redesign phase last up until now?
  • Chapter 2.4.8.: It would be good to provide some more information about actual redesign requests from experts. Was it only default values, less clicks?
  • Table 2: There bottom border in the first column is missing

I am looking forward to have this manuscript published. I believe that the community will benefit from the insights of the authors.
